# Role of Rac1 in p53-Related Proliferation and Drug Sensitivity in Multiple Myeloma [note 1]

**DOI:** 10.3390/cancers17030461

**Published:** 2025-01-29

**Authors:** Ikuko Matsumura, Tsukasa Oda, Tetsuhiro Kasamatsu, Yuki Murakami, Rei Ishihara, Ayane Ohmori, Akira Matsumoto, Nanami Gotoh, Nobuhiko Kobayashi, Yuri Miyazawa, Yoshiyuki Ogawa, Akihiko Yokohama, Nobuo Sasaki, Takayuki Saitoh, Hiroshi Handa

**Affiliations:** 1Hematology, Graduate School of Medicine, Gunma University, Maebashi 371-8511, Gunma, Japan; 2Mucosal Ecosystem Design, Institute for Molecular and Cellular Regulation, Gunma University, Maebashi 371-8511, Gunma, Japan; 3Faculty of Medical Technology and Clinical Engineering, Gunma University of Health and Welfare, Maebashi 371-0823, Gunma, Japan; 4Department of Laboratory Sciences, Graduate School of Health Sciences, Gunma University, Maebashi 371-8511, Gunma, Japan; 5Division of Blood Transfusion, Gunma University Hospital, Maebashi 371-8511, Gunma, Japan

**Keywords:** multiple myeloma, Rho-GTPase signaling, Ras-related C3 botulinus toxin substrate 1 (Rac1), p53, chemosensitivity

## Abstract

Multiple myeloma (MM) is characterized by monoclonal plasma cell proliferation in the bone marrow. p53 plays a critical role in MM and the 17p deletion is associated with poor clinical outcomes. Rho-GTPase signaling, which includes Ras-related C3 botulinum toxin substrate 1 (Rac1) as a major regulator, is involved in cancer progression and chemoresistance; however, the role of Rac1 in MM remains elusive. This study was conducted to evaluate the role of Rac1 in MM. The results revealed that Rac1 affected the survival of human MM cell lines (HMCLs) independent of p53 status. High *RAC1* mRNA expression in the intramedullary plasma cells of patients with NDMM was associated with poor overall survival. The Rac1 inhibitor intensified the effect of cereblon modulators on HMCLs. This study presents new insights into developing novel therapies targeting the Rac1 pathway, potentially improving the prognosis of patients with MM, including those with Wild Type p53 deficiency.

## 1. Introduction

In this work, the study presented in [1] is expanded upon. Multiple myeloma (MM) is a malignancy characterized by monoclonal plasma cell proliferation in the bone marrow (BM), leading to the production of excessive monoclonal immunoglobulins [2,3]. The p53 gene, a tumor suppressor gene on chromosome 17p 13.1, plays a critical role in several tumor types, including MM. The 17p deletion, which is associated with poor clinical outcomes, occurs in 7–10% of patients with newly diagnosed MM (NDMM) [4,5,6]. However, effective treatments for this patient group remain unestablished, highlighting the imperative of discovering new molecular targets to develop novel therapies.

Rho-GTPases are a family of small G proteins of the Ras superfamily, with the Ras homolog family member A (RhoA), RAS-related C3 botulinus toxin substrate 1 (Rac1), and cell division control protein 42 (Cdc42) as major regulators. The Rho-GTPase family regulates the dynamics of the actin cytoskeleton, as well as cell proliferation, differentiation, motility, adhesion, survival, and secretion [7]. Rho-GTPase signaling is involved in cancer progression, dissemination, and chemoresistance, including in hematological tumors [8]. Rho-GTPase inhibitors, such as the Rac1 inhibitor NSC23766, suppress the proliferation of human MM cell lines (HMCLs) [9]. Furthermore, p53 deficiency increases the Rac1 activity in both B- and T-cell lines [10]. However, the role of Rac1 in MM progression and its relationship with p53 remains elusive. Additionally, the relationship between Rac1 and chemoresistance in MM has not yet been elucidated. Therefore, this study was conducted to evaluate the relationship between Rac1 and p53 in MM progression, as well as the role of Rac1 in drug sensitivity in MM.

## 2. Materials and Methods

### 2.1. Cell Lines

Three HMCLs, namely KMS11, KMS26, and MM.1S, with tumor protein p53 (*TP53*) in the deficient, mutated, and wild statuses, respectively, were used in this study. KMS11 and KMS26 cells were provided by Dr. Takemi Otsuki (Kawasaki Medical School, Okayama, Japan), whereas MM.1S cells were obtained from Deutsche Sammlung von Mikroorganismen und Zellkulturen (Braunschweig, Germany). HMCLs were cultured in RPMI-1640 medium (Sigma-Aldrich, St Louis, MO, USA) supplemented with 10% fetal bovine serum at 37 °C in 5% CO_2_. Cell proliferation was determined using a water-soluble tetrazolium-8 (WST-8) assay (Dojindo Laboratories, Kumamoto, Japan).

### 2.2. Patients

In total, 114 patients with NDMM and 70 patients with monoclonal gammopathy of undetermined significance between April 2015 and January 2021 were enrolled in this study, excluding patients with amyloidosis. Fifteen patients were enrolled as controls between May 2014 and January 2015, including individuals with lymphoma without bone marrow (BM) infiltration, acute myeloid leukemia in complete remission, immune thrombocytopenic purpura, aplastic anemia, and idiopathic cytopenia of unknown significance. High-risk cytogenetics was defined as the presence of more than one of the following deletions: (17p), t(4;14), t(14;16), t(8;14), and t(14;20). Plasma cells were purified from BM mononuclear cells using an anti-CD138 antibody conjugated with phycoerythrin (PE) (Beckman-Coulter, Brea, CA, USA) and an EasySep PE positive selection kit containing anti-PE antibody conjugated with micro-magnetic beads (STEMCELL Technologies, Vancouver, BC, Canada). Overall survival (OS) was defined as the interval from the date of BM aspiration for diagnosis to death, and progression-free survival (PFS) was defined as the interval from the date of therapy initiation to disease progression. Two patients with smoldering multiple myeloma treated in a clinical trial were excluded from the OS and PFS analyses.

### 2.3. Reagents

In this study, several inhibitors and compounds were used, including the Rac1 inhibitor 1A-116, pomalidomide, iberdomide (Selleck Chemicals, Huston, TX, USA), mouse double minute 2 (Mdm2) inhibitor nutlin-3 (Cayman Chemical, Ann Arbor, MI, USA), bortezomib, and lenalidomide (FUJIFILM Wako Pure Chemical Corporation, Osaka, Japan).

### 2.4. Expression of Wild-Type p53 Using Tet-On System

KMS11 and KMS26 expressing doxycycline-inducible p53 (KMS11/Tet-on p53 and KMS26/Tet-on p53, respectively) were obtained through infection with the pseudo-lentivirus and selection with 1 µg/mL puromycin (Sigma-Aldrich, St Louis, MO, USA). The induction of p53 expression was achieved using 1 µg/mL doxycycline (TaKaRa Bio, Kyoto, Japan) [11]. The detailed methods are provided in the Appendix A.

### 2.5. Gene Knockdown

Pseudo-lentivirus producing short hairpin RNA targeting *TP53* (sh*TP53*) or sh*RAC1* were prepared as described in Section 2.4 and used to infect MM.1S cells for *TP53* knockdown or KMS11 and KMS26 cells for *RAC1* knockdown. Sh green fluorescent protein (*GFP*) was prepared and used to infect KMS11 and KMS26 cells as controls. The knockdown cells were obtained through selection with 10 µg/mL blasticidin S hydrochloride (FUJIFILM Wako Pure Chemical) three days after virus infection. mRNA expression was determined seven days post-infection, and cell viability was determined using a WST-8 assay eight days post-infection. The target *TP53*, *RAC1*, and *GFP* sequences were 5′-GACUCCAGUGGUAAUCUAC-3′, 5′-CCCTACTGTCTTTGACAATTA-3′, and 5′-GCAAGCUGACCCUGAAGUUCA-3′, respectively.

### 2.6. Isolation of Nucleic Acids and RNA Expression Analysis Using PCR

RNA was extracted from HMCLs and patient BM plasma cells using the RNeasy Mini Kit (Qiagen, Hilden, Germany) or mirVana miRNA Isolation Kit (Ambion, Austin, TX, USA). The RNA quantity and quality were measured using BioSpec-nano (SHIMADZU, Kyoto, Japan). cDNA was synthesized using the PrimeScript™ RT Reagent Kit with the gDNA Eraser (TaKaRa Bio, Shiga, Japan). The levels of *TP53*, cyclin-dependent kinase inhibitor 1A (*CDKN1A*), *MDM2*, and *RAC1* mRNA were determined via real-time PCR using the Power Up SYBR Green PCR Master Mix (Applied Biosystems, Foster City, CA, USA). Actin beta (*ACTB*) served as an endogenous control gene, and MM.1S was used as a reference sample. Primers used for reverse transcription quantitative PCR (RT-qPCR) are listed in Appendix A.

### 2.7. Western Blotting Analysis

p53, p21, Mdm2, and Rac1 protein expression levels were determined using western blotting. The cells were lysed in sodium dodecyl sulfate (SDS) lysis buffer (62.5 mM Tris [pH, 6.8], 2% SDS, and 10% glycerol) (made in house) and sonicated. β-mercaptoethanol and bromophenol blue were added to the lysates at concentrations of 5%. The lysates were boiled and used as whole-cell lysates. Equal amounts of protein (8–20 μg) were subjected to electrophoresis using a 12% precast polyacrylamide gel (Bio-Rad Laboratories, Hercules, CA, USA). The separated proteins were transferred onto a polyvinylidene fluoride membrane (Immobilon-P Transfer Membrane, Merck KGaA, Darmstadt, Germany) using a semi-dry transfer apparatus (AE-6688 HorizeBLOT 4M; ATTO, Tokyo, Japan) or a submarine transfer apparatus (Criterion Blotter, Bio-Rad Laboratories, Hercules, CA, USA). Western blotting was performed according to standard procedures. The expression of proteins was detected by an HRP-conjugated secondary antibody with ECL or by an AP-conjugated secondary antibody with BCIP and NBT, with ACTB serving as a loading control. The band intensities were measured using the ImageJ software v.1.54d (National Institutes of Health, Bethesda, MD, USA). The antibodies used for western blotting are listed in Appendix A.

### 2.8. Flow Cytometry Analysis

For the proliferation assay, HMCLs were treated with 1A-116 and labeled with the 5-ethynyl-2′-deoxyuridine (EdU) staining proliferation kit (Abcam, Cambridge, UK) 24 h post-treatment, according to the manufacturer’s protocol. The cells were cultured in a medium containing 20 µM EdU for 4 h and stained with iFluor 488 azide (Abcam, Cambridge, UK). For the apoptosis assay, the HMCLs were collected 24 h, 48 h, and 72h after 1A-116 treatment and stained with Annexin V binding buffer (10 mM 4-(2-Hydroxyethyl)-1-piperazineethanesulfonic acid [pH, 7.4], 140 mM NaCl, and 2.5 mM CaCl2) (made in house), 7-aminoactinomycin D (AAD), and Annexin V conjugated to fluorescein isothiocyanate (BioLegend, San Diego, CA, USA). The proportion of EdU-positive and apoptotic cells was determined using a BD FACSCanto™ II flow cytometer (BD Biosciences, Franklin Lakes, NJ, USA). Flow cytometry was performed according to standard procedures. Data were analyzed using Flowing Software v.2.5.1 (Turku Bioscience, Turku, Finland).

### 2.9. RNA Sequencing

Whole transcriptome analysis was performed using the NextSeq 500 instrument (Illumina, San Diego, CA, USA) with a NextSeq 500/550 high-output kit v2.5 (75 cycles) (FC-404-2005; Illumina) [12]. Gene ontology (GO) analysis was performed using Metascape software (v.3.5). The detailed methods are provided in the Appendix A.

### 2.10. Statistical Analysis

All calculations were performed using the EZR software package (version 1.54) [13]. Results with *p*-values < 0.05 were considered statistically significant. The RT-qPCR data, viable cell rates, and positive cell rates were divided into control and test groups and analyzed using Student’s *t*-test. Continuous values for three or more groups were evaluated using a one-way analysis of variance or Kruskal–Wallis tests. Frequencies were evaluated using Fisher’s exact test, and continuous values for comparing patient backgrounds were evaluated using the Mann–Whitney U test. The OS and PFS were evaluated using log-rank tests for univariate analysis. The Cox regression hazard model was used for multivariate analysis.

## 3. Results

### 3.1. Gene Ontology Analysis of p53, p21, and Mdm2 Protein Expression and Proliferation in Tet-On p53 HMCLs

p53, p21, and Mdm2 protein expression levels after 24 h of doxycycline exposure were determined using western blotting. The expressions of p53 and p21 were observed. Mdm2 expression was upregulated in KMS11/Tet-on p53 cells, and both p21 and Mdm2 expressions were observed in KMS26/Tet-on p53 cells upon WT p53 induction. The proliferation rate of KMS11/Tet-on p53 cells was unaffected by p53 induction, whereas that of KMS26/Tet-on p53 cells was significantly reduced (Figure 1A,B).

Gene expression profiles were compared using RNA sequencing to elucidate the determinants of the differing responses to WT p53 induction in the HMCLs. Whole transcriptome analysis revealed that 4896 genes were more highly expressed in KMS11 cells than in KMS26 cells. Of these, 3000 genes were selected for GO analysis owing to their significant differences, with an adjusted *p*-value. GO analysis demonstrated that Rho-GTPase signaling was enhanced in KMS11 cells compared to that in KMS26 cells (Figure 1C).

### 3.2. Expression of RAC1 in Multiple Myeloma Patient Samples and HMCLs

Among the three major Rho-GTPases—RhoA, Rac1, and Cdc42—Rac1 was selected because it has been previously associated with p53 in a study on lymphoma [10]. The *RAC1* mRNA levels in purified BM plasma cells, determined using qRT-PCR, were significantly higher in patients with NDMM than in the controls (*p* < 0.01) (Figure 2A). In patients with NDMM, the *RAC1* mRNA levels in myeloma cells did not differ according to the karyotype, International Staging System (ISS), or Revised-ISS (Figure 2B–D).

*RAC1* mRNA and protein expression were detected in KMS11, KMS26, and MM.1S cells. However, the *RAC1* mRNA levels in KMS11 and KMS26 cells were higher than those in the controls (Appendix A). WT p53 induction did not alter *RAC1* mRNA or Rac1 protein levels in KMS11/Tet-on p53 and KMS26/Tet-on p53 cells at 24 h. In MM.1S cells, *RAC1* mRNA and Rac1 protein expression at 24 h was independent of the p53 status (Figure 2E,F). Similarly, Rac1 protein expression at 48 h and 72 h was independent of the p53 status (Appendix A).

### 3.3. MM Cell Proliferation and Apoptosis After Treatment with 1A-116

The 50% inhibitory concentrations (IC50) of 1A-116 at 72 h, determined using a WST-8 assay and ImageJ software, were 49.5 µM in KMS11 and 111.6 µM in KMS26, respectively (Appendix A). Treatment with the Rac1 inhibitor 1A-116 (50 μM) resulted in significantly reduced cell survival in both KMS11 and KMS26 cells at 72 h, as detected by the WST8 assay. This effect was more pronounced in KMS11 cells than in KMS26 cells. 1A-116 also significantly reduced the survival of MM.1S cells, both with p53 induction by nutlin-3 (1 µM) and *TP53* knockdown by Sh*TP53* at 72 h, although the knockdown of WT *TP53* or nutlin-3 alone did not affect survival (Figure 3A). In KMS11/Tet-on p53 and KMS26/Tet-on p53 cells, treatment with 1A-116 did not increase p53, p21, or Mdm2 protein expression. In MM.1S cells cotreated with nutlin-3 and 1A-116, 1A-116 did not increase p53, p21, or Mdm2 protein expression (Figure 3B). Similarly, 1A-116 did not increase p53 and p21 at 48 h and 72 h (Appendix A).

EdU and Annexin V assays were performed to determine whether Rac1 inhibition suppresses cell proliferation or induces cell death. 1A-116 treatment significantly reduced EdU incorporation in all three HMCLs, indicating that Rac1 inhibition arrested the cell cycle (Figure 3C); however, the effect was less pronounced in KMS11 cells. Additionally, the proportion of apoptosis and dead cells, identified by Annexin V-positivity and 7-AAD incorporation, was significantly increased with 1A-116 treatment in all three HMCLs (Figure 3D). These results suggest that Rac1 inhibition can arrest the cell cycle and induce cell death independent of the p53 pathway.

### 3.4. Cell Survival in RAC1 Knockdown MM Cells and Transcriptome Analysis

To further elucidate the role of Rac1 in MM cell survival, *RAC1* knockdown was induced in KMS11 and KMS26 cells using shRNA. *RAC1* knockdown significantly reduced the survival of KMS11 cells but not that of KMS26 cells. *TP53*, *CDKN1A*, and *MDM2* mRNA levels remained unchanged by *RAC1* knockdown, reinforcing the notion that the reduction in MM cell survival via Rac1 inhibition is independent of the p53 status (Figure 4A,B).

### 3.5. Effect of Rac1 on Sensitivity Against Cereblon Modulators and Transcriptome Analysis

Given that Rac1 inhibition reduced MM cell survival regardless of the p53 status, we examined whether it could increase MM cell sensitivity to existing drugs, including cereblon (CRBN) modulators and proteasome inhibitors, for the development of new therapeutic strategies. To evaluate the synergic effect of existing drugs, we used 1A-116 at a concentration of 25 µM that did not affect KMS11 and KMS26 survival (Appendix A). The combination of 1A-116 (25 µM) with CRBN modulators—lenalidomide, pomalidomide, and iberdomide—significantly decreased cell survival in KMS11 and KMS26 cells at 72 h compared to that observed with the CRBN modulators alone. However, 1A-116 did not exhibit any synergistic additive effect combined with bortezomib in all three HMCLs after treatment for 24 h (Figure 5A,B).

### 3.6. Alteration of Gene Expression Profile by Rac1 Inhibition

To gain a deeper understanding of the role of Rac1 inhibition, we analyzed the gene expression profiles of MM cells using RNA sequencing following treatment with 1A-116 (25 µM) and *RAC1* knockdown.

In KMS11 cells, 137 and 54 genes were significantly upregulated and downregulated, respectively, by 1A-116 (25 µM). In KMS26 cells, 109 and 111 genes were significantly upregulated and downregulated, respectively, by the same treatment. The GO analysis demonstrated that genes associated with cholesterol synthesis disorders were upregulated, whereas those related to ADP metabolic process and response to oxygen levels were downregulated in both KMS11 and KMS26. In KMS11 cells, genes associated with regulated exocytosis was upregulated (Figure 6A,B).

In the knockdown experiment, 1421 and 1023 genes were significantly upregulated and downregulated, respectively, by *RAC1* knockdown in the KMS11 cells. Conversely, 1352 and 1288 genes were significantly upregulated and downregulated, respectively, by *RAC1* knockdown in the KMS26 cells. The GO analysis revealed that genes associated with signaling by Rho GTPases, micro GTPases, and Rho-related BTB domain-containing protein 3 was upregulated, whereas those related to cell cycle, membrane trafficking, and membrane organization were downregulated (Figure 6C,D).

### 3.7. Prognosis of Patients with MM

No significant differences were observed in the characteristics of patients with NDMM between those with high *RAC1* mRNA expression (above the median value) and those with low expression (Table 1). Patients with high *RAC1* mRNA expression had a significantly shorter median time of OS (4.3 years vs. not reached; *p* = 0.01), although the PFS did not differ significantly between the two groups (2.0 years vs. 3.1 years; *p* = 0.21) (Figure 7A). Multivariate analysis identified both ASCT and *RAC1* mRNA expression as independent prognostic factors for OS (ASCT performed: hazard ratio [HR], 0.409; *p* = 0.04; high *RAC1* mRNA expression: HR, 2.211; *p* = 0.02). However, ASCT was the only independent prognostic factor for PFS (ASCT performed: HR, 0.405; *p* = 0.01) (Table 2).

Given that ASCT has demonstrated efficacy in improving prognosis in MM [14], OS and PFS were analyzed separately in the groups with and without ASCT. Among patients who underwent ASCT, both OS and PFS were significantly worse for those with high *RAC1* mRNA expression compared to the values for those with low *RAC1* expression (OS: 5.1 years vs. not reached, *p* = 0.02; PFS: 2.7 years vs. not reached, *p* = 0.01) (Figure 7B). Similarly, among patients who did not undergo ASCT, OS was significantly shorter for those with high *RAC1* mRNA expression (OS: 4.3 years vs. 7.3 years; *p* = 0.048); however, PFS did not differ significantly (Figure 7C).

## 4. Discussion

The results of this study demonstrated that Rac1 inhibition reduced MM cell proliferation, independent of the p53 status, by suppressing the cell cycle and inducing apoptosis. Furthermore, Rac1 inhibition enhanced sensitivity to CRBN modulators. *RAC1* mRNA levels in purified BM plasma cells were significantly higher in patients with NDMM than in the controls, and high *RAC1* mRNA expression in MM cells was associated with poor OS outcomes, regardless of ASCT, with a more pronounced effect in patients who underwent ASCT.

The Rac1 inhibitor 1A-116 significantly reduced cell proliferation in KMS11, KMS26, and MM.1S cells with *TP53* deficiency, mutation, and WT knockdown, respectively. These findings indicated that Rac1 inhibition affected MM cell proliferation regardless of the p53 status, consistent with a report on lymphoma suggesting that Rac1 targeting may affect cell growth independent of the p53 status [10]. The effect of 1A-116 was more pronounced in KMS11 cells than in KMS26 cells, suggesting that KMS11 cells, resistant to p53 activation, may be more dependent on Rac1 for survival and proliferation.

1A-116 did not increase the p53, p21, or Mdm2 protein expression in KMS11 and KMS26 cells. Similarly, *RAC1* knockdown did not upregulate *TP53*, *CDKN1A*, or *MDM2* mRNA levels in the KMS11 and KMS26 cells. These results suggested that the effect of Rac1 inhibition was not exerted through the p53 pathway activation, suggesting that MM may progress when Rac1 oncogenic activity exceeds the tumor suppressor effect of p53.

Furthermore, 1A-116 suppressed the cell cycle in HMCLs, and GO analysis revealed that *RAC1* knockdown suppressed genes related to the cell cycle in KMS11 and KMS26 cells. These results are consistent with previous reports on the relationship between Rac1 and the cell cycle. For example, cyclin D1 expression, a key event in G1 phase progression, is induced by Rac1 in mammary epithelial cells [15]. Rac1 also plays an essential role in the activation of irradiation-induced extracellular signal-regulated kinases 1 and 2 signaling and the subsequent G2/M checkpoint response in breast cancer cells [16].

Our results revealed markedly induced apoptosis in KMS11 cells by the Rac1 inhibitor, which is consistent with previous results indicating that Rac1 is involved in a p53-independent apoptotic pathway in human lymphoma cells [10]. Rac1 inhibits apoptosis in p53-deficient lymphoma cells by stimulating B-cell lymphoma 2 (Bcl-2)-associated death promoter (Bad) phosphorylation at Ser75 via protein kinase A and not protein kinase B in response to DNA-damaging chemotherapies [17].

Rac1 is also identified as a major mediator of chemoresistance in malignancy [18]. In chronic lymphocytic leukemia, Rac1 and its guanine nucleotide exchange factor T-cell lymphoma invasion and metastasis 1 are important for proliferation and chemoresistance to fludarabine, a DNA intercalating purine analog [19]. One primary mechanism by which Rac1 induces resistance to chemotherapy may be through apoptosis regulation. Rac1 inhibition enhances leukemia cell sensitivity to etoposide-induced apoptosis [20]. Constitutively active Rac1 stimulates bad phosphorylation, suppressing drug-induced caspase activation and apoptosis in human lymphoma cells [17]. Moreover, in lymphoblastic cell lines, Rac1 binds to BCL-2, stabilizing its anti-apoptotic properties and contributing to chemoresistance [21].

The role of Rac1 in chemoresistance in MM remains elusive. Proteasome inhibitors (bortezomib) and CRBN modulators (lenalidomide) are crucial for MM treatment. However, most patients experience relapse and develop multidrug resistance owing to prolonged exposure [22]. In this study, Rac1 inhibition increased HMCL sensitivity to CRBN modulators. Lenalidomide, pomalidomide, and iberdomide mediate their anti-myeloma activities via CRBN, IKAROS family zinc finger (IKZF)1, and IKZF3 [23,24,25]. CRBN modulators induce several downstream changes, such as the inactivation of nuclear factor-κB and the activation of caspase 8 [26,27]. Although bortezomib is associated with apoptosis in MM [28], Rac1 inhibition did not enhance sensitivity to bortezomib. While apoptosis pathways may be associated with CRBN modulator sensitivity in HMCLs, factors beyond apoptosis may influence drug sensitivity.

GO analysis revealed that 1A-116 (25 µM) upregulated the expression of genes associated with cholesterol and exocytosis while downregulating those of genes related response to oxygen levels. Exosomes play a role in MM drug resistance [29]. Sortilin 1/lysosomal associated membrane protein 2-mediated extracellular vesicle secretion and cell adhesion are linked to lenalidomide resistance in MM [22]. Furthermore, advanced cancers are characterized by the generation of a hypoxic environment and the activation of its main effector, HIF-1 [30]. In MM, hypoxic responses are crucial for maintaining cellular homeostasis, inducing traits adapted to hypoxia and leading to drug resistance [31]. RhoA and Rac1 have been demonstrated to stabilize the HIF-1α protein, suggesting the possible cross-talk between small G-proteins and HIF pathways in hepatocellular carcinoma [32]. However, the role of Rac1 in hypoxia in MM remains unclear.

Our results of cotreated CRBN modulators and 1A-116 (25µM) suggested that Rac1 inhibition may enhance CRBN modulator sensitivity. Further investigations are required to elucidate how Rac1 inhibition affects CRBN sensitivity. Moreover, exploring the relationship between Rac1 and exosomes, membranes, and hypoxia responses could enhance CRBN modulator efficacy.

The *RAC1* mRNA levels were significantly higher in the MM cells obtained from patients with NDMM than in the control patients, consistent with previous results [33]. Subsequently, patient survival was analyzed according to the *RAC1* mRNA levels to assess its clinical significance. The significantly shorter OS in patients with high *RAC1* mRNA expression indicates the significant role of Rac1 in MM treatment, supporting our in vitro findings of the role of Rac1 in MM cell survival and drug sensitivity. Given the efficacy of Rac1 inhibition in breast cancer and glioma in vivo models [34,35], targeting Rac1 could help circumvent drug resistance, specifically with CRBN modulators and p53 inactivation. Further studies, including in vivo models, are required to elucidate the specific roles of Rac1 in MM and facilitate the development of novel therapies targeting the Rac1 pathway.

## 5. Conclusions

Our results revealed that Rac1 affected HMCL survival regardless of the p53 status and is associated with CRBN moderator sensitivity. This study presents the first endeavor to associate high *RAC1* mRNA expression in the intramedullary plasma cells of patients with NDMM with a worse prognosis. Overall, our study provided novel insights crucial for developing new therapies targeting the Rac1 pathway and improving the prognosis of patients with MM, including those with p53 deficiency and mutation.

## Figures and Tables

**Figure 1 cancers-17-00461-f001:**
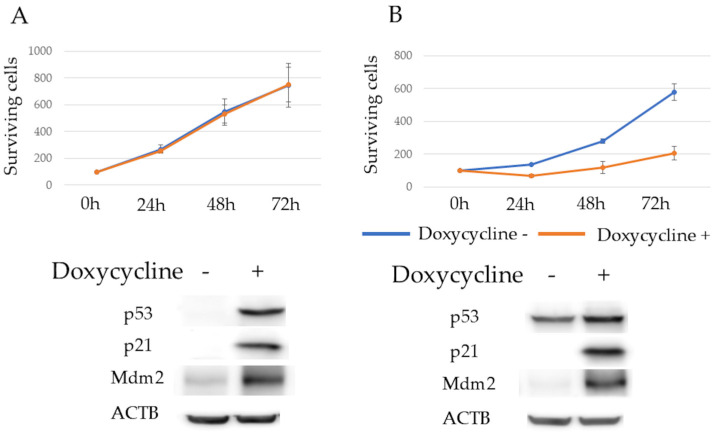
Surviving cells of KMS11 were unaffected by p53 induction, but the Rho-GTPase signal was enhanced in KMS11 cells compared to that in KMS26 cells. (**A**) The surviving cells and protein expression of actin beta (ACTB), p53, p21 and mouse double minute 2 (Mdm2) at 24 h after treatment with 1 μg/mL of doxycycline in KMS11 expressing doxycycline-inducible p53 (KMS11/Tet-on p53) and (**B**) KMS26 (KMS26/Tet- on p53). Blue, no doxycycline; orange, with 1 μg/mL of doxycycline. Error bars represent the standard deviation from four experiments. (**C**) Gene ontology analysis revealed that genes were more highly expressed in KMS11 cells than in KMS26 cells. Original images of western blotting can be found in Appendix A.

**Figure 2 cancers-17-00461-f002:**
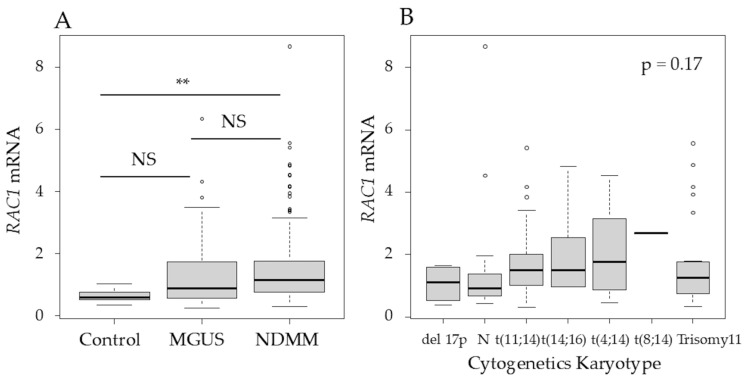
RAS-related C3 botulinus toxin substrate 1 (*RAC1*) mRNA levels in purified bone marrow (BM) plasma cells were higher in patients with newly diagnosed MM (NDMM) than in controls. (**A**) *RAC1* mRNA levels in purified BM plasma cells in controls and patients with monoclonal gammopathy of undetermined significance (MGUS) or NDMM. The Y axis represents the delta–delta Ct value, with actin beta (*ACTB*) serving as an endogenous control gene and MM.1S as the reference sample. (**B**) *RAC1* mRNA levels in different cytogenetic abnormalities, (**C**) International Staging System (ISS), and (**D**) Revised-ISS. (**E**) *RAC1* mRNA and (**F**) Rac1 expression of human myeloma cell lines based on p53 status, 24 h after treatment. KD, knockdown; NS, not significant; **, *p* < 0.01. Original images of western blotting can be found in Appendix A.

**Figure 3 cancers-17-00461-f003:**
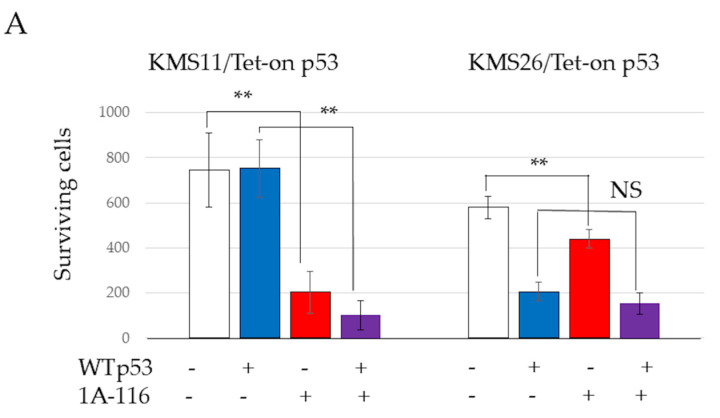
RAS-related C3 botulinus toxin substrate 1 (Rac1) affected human myeloma cell lines (HMCLs) survival, cell cycle, and apoptosis independent of p53 status. (**A**) HMCL living cells at 72 h after treatment with 50 μM of Rac1 inhibitor 1A-116. White color represents the control and blue indicates the Wild Type (WT). p53 expression is higher than that in the control group. The red color indicates the control treated with 1A-116. The purple color represents the WTp53 expression, which is higher than that in the control group treated with 1A-116. (**B**) Protein expression of p53, p21, mouse double minute 2 (Mdm2), and actin beta (ACTB) at 24 h after treatment with 1A-116 based on p53 status. (**C**) 5-ethynyl-2′-deoxyuridine (EdU) positive cells of HMCLs treated with 1A-116 (50 µM). (**D**) HMCL cells undergoing apoptosis treated with 1A-116 (0, 25, and 50 µM). The dot plot shows results at 48 h. White, treatment with 1A-116 0 µM; pink, 25 µM; red, 50 µM. The error bars indicate the standard deviation across more than three experiments. NS, not significant; *, *p* < 0.05; **, *p* < 0.01. Original images of western blotting can be found in Appendix A.

**Figure 4 cancers-17-00461-f004:**
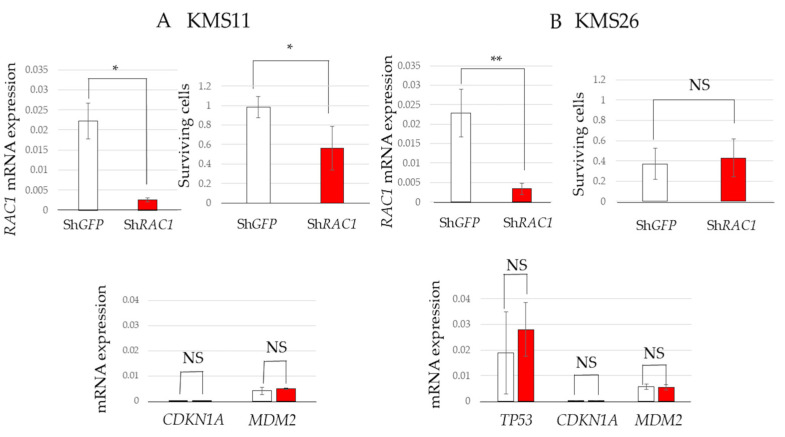
Short hairpin RNA targeting RAS-related C3 botulinus toxin substrate 1 (Sh*RAC1*) significantly reduced the survival of KMS11 cells. Living cells of HMCLs at eight days after virus infection and mRNA expression of *RAC1*, tumor protein p53 (*TP53*), cyclin-dependent kinase inhibitor 1A (*CDKN1A*), and mouse double minute 2 (*MDM2*) at seven days after virus infection in (**A**) KMS11 and (**B**) KMS26. Error bars show the standard deviation within the three experiments. White, Sh green fluorescent protein (*GFP*); red, Sh*RAC1*; NS, not significant; *, *p* < 0.05; **, *p* < 0.01.

**Figure 5 cancers-17-00461-f005:**
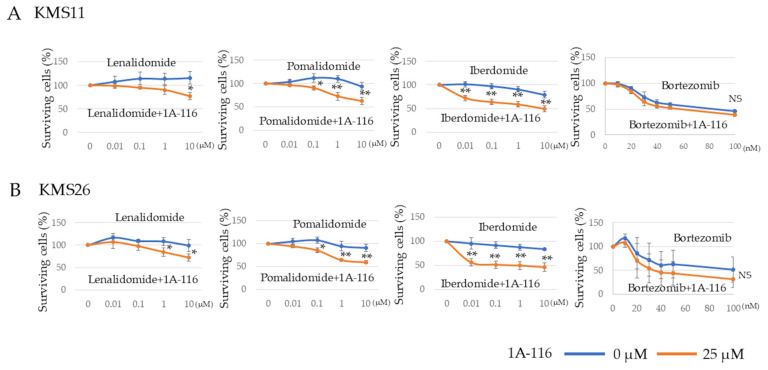
RAS-related C3 botulinus toxin substrate 1 (Rac1) was implicated in the sensitivity of human myeloma cell lines to cereblon modulator. We used 1A-116 at a concentration of 25 µM that did not affect KMS11 and KMS26 survival. Living cells of HMCLs, (**A**) KMS11 and (**B**) KMS26 after treatment with lenalidomide, pomalidomide, iberdomide, and bortezomib, respectively. Error bars show the standard deviation across three experiments. Blue, no Rac1 inhibitor 1A-116; orange, with 1A-116 (25 µM). NS, not significant; *, *p* < 0.05; **, *p* < 0.01.

**Figure 6 cancers-17-00461-f006:**
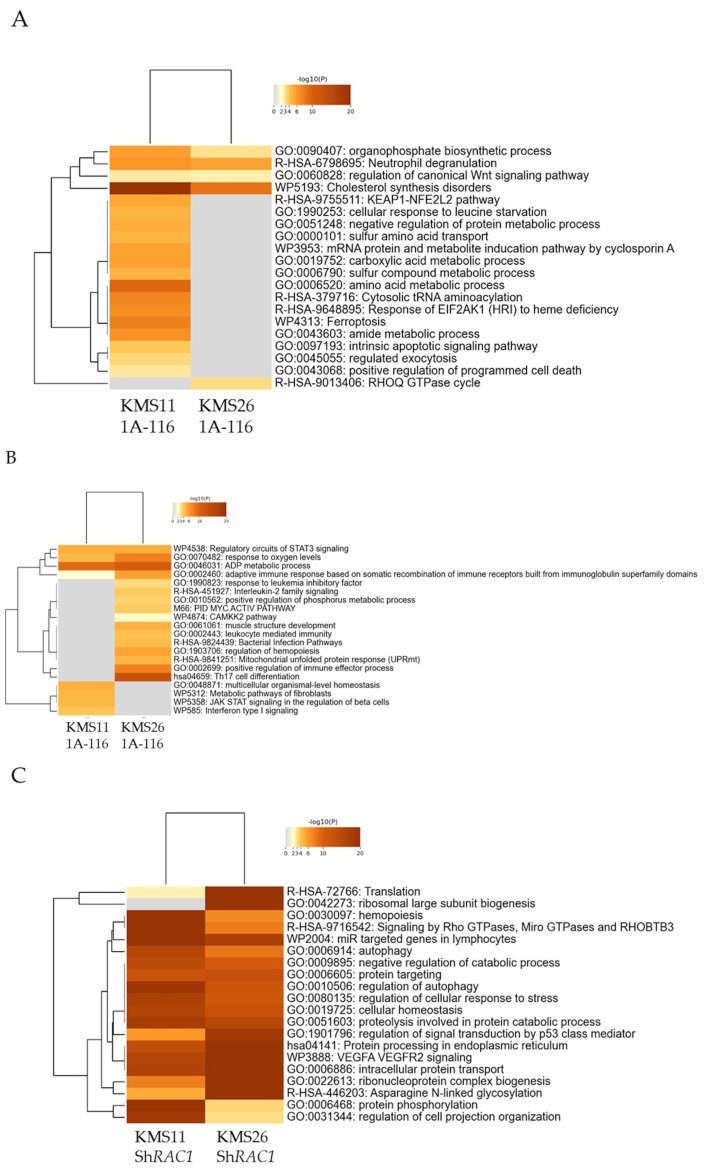
Gene ontology (GO) analysis of the gene expression profiles of human myeloma cell lines following treatment with 1A-116 or RAS-related C3 botulinus toxin substrate 1 (*RAC1*) knockdown. The classification is based on significantly upregulated genes by 25 μM of 1A-116 (**A**) and downregulated by same treatment (**B**) in KMS11 and KMS26. The classification is based on significantly upregulated genes by *RAC1* knockdown (**C**) and downregulated by *RAC1* knockdown (**D**) in KMS11 and KMS26.

**Figure 7 cancers-17-00461-f007:**
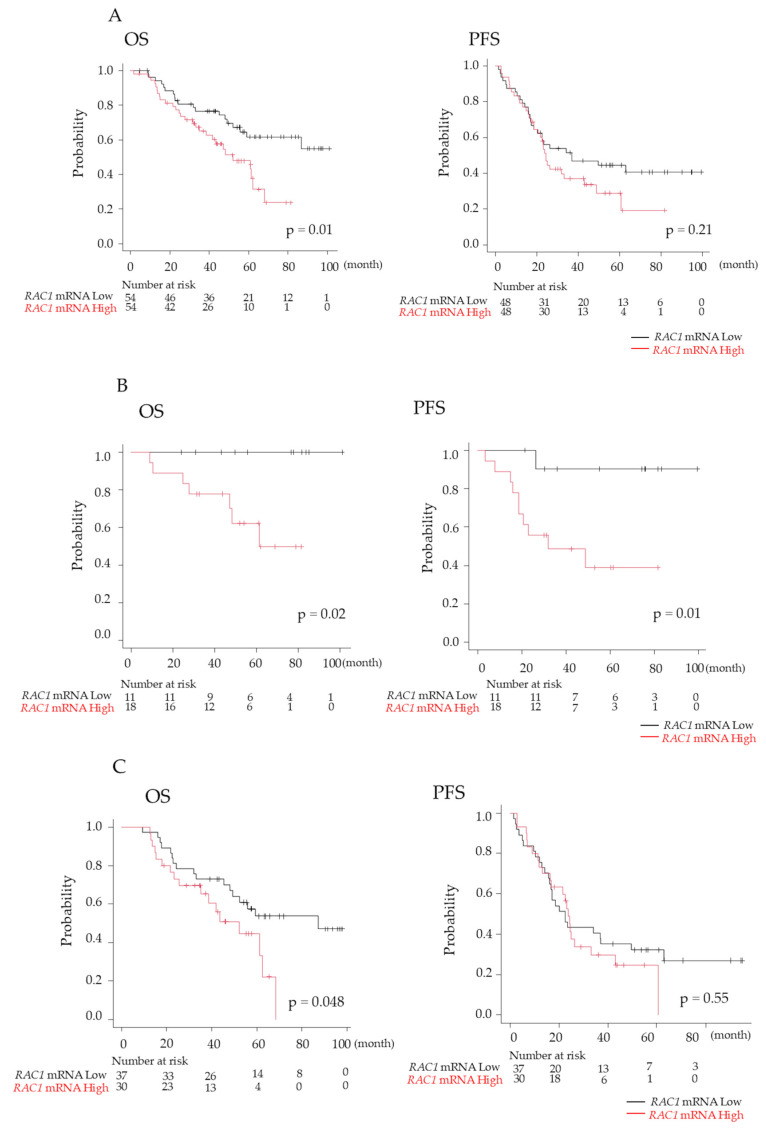
Prognosis of patients with newly diagnosed multiple myeloma (NDMM). (**A**) Overall survival (OS) and progression-free survival (PFS) were compared between high and low RAS-related C3 botulinus toxin substrate 1 (*RAC1*) gene expression of plasma cells in bone marrow (BM) in patients with NDMM. OS and PFS were analyzed separately in patients with NDMM groups with (**B**) and without (**C**) autologous stem cell transplantation. Black line, *RAC1* mRNA levels of plasma cells in BM < 1.17; red line, *RAC1* mRNA levels > 1.17.

**Table 1 cancers-17-00461-t001:** Characteristics of patients with multiple myeloma with high and low *RAC1* mRNA expression.

	*RAC1* High (*n* = 57)	*RAC1* Low (*n* = 57)	*p*-Value
Age	71 (41–87)	71 (42–86)	1.00
Sex			0.70
Male	24 (42.1%)	21 (36.8%)	
Female	33 (57.9%)	36 (63.2%)	
ALB	3.45 (2.0–4.8)	3.60 (2.2–4.6)	0.73
Hb	10.65 (5.9–15.7)	10.10 (6.3–14.6)	0.17
LDH	177 (92–353)	169 (99–497)	0.78
β2MG	3.8 (1.4–27.0)	4.4 (1.8–29.1)	0.21
IgH			0.43
IgG	37 (66.1%)	32 (56.1%)	
IgA	12 (21.4%)	16 (28.1%)	
IgD	0 (0.0%)	2 (3.5%)	
IgE	0 (0.0%)	1 (1.8%)	
BJ	7 (12.5%)	5 (8.8%)	
Non	0 (0.0%)	1 (1.8%)	
IgL			0.16
κ	35 (62.5%)	27 (47.4%)	
λ	21 (37.5%)	29 (50.9%)	
Non	0 (0.0%)	1 (1.8%)	
Cytogenetic Risk			0.28
Standard	43 (79.6%)	40 (70.2%)	
High	11 (20.4%)	17 (29.8%)	
ISS			0.59
1	15 (27.3%)	11 (21.2%)	
2	24 (43.6%)	21 (40.4%)	
3	16 (29.1%)	20 (38.5%)	
R-ISS			0.49
1	12 (22.6%)	7 (13.2%)	
2	36 (67.9%)	40 (75.5%)	
3	5 (9.4%)	6 (11.3%)	
ASCT			0.18
Yes	11 (22.9%)	18 (37.5%)	
No	37 (77.1%)	30 (62.5%)	

Patients for whom data were unavailable were excluded from the analysis. BJ, Bence-Jones type; Non, non-secretory type; ISS, International Staging System; R-ISS, Revised International Staging System; ASCT, autologous stem cell transplantation; RAC1, Ras-related C3 botulinum toxin substrate 1; ALB, albumin; LDH, lactate dehydrogenase; β2MG, beta-2 microglobulin.

**Table 2 cancers-17-00461-t002:** Multivariate analysis of predisposing factors for overall survival and progression-free survival in patients with MM.

		OS Hazard Ratio (95% CI)	*p*-Value	PFS Hazard Ratio (95% CI)	*p*-Value
R-ISS	≥2	1.751	0.30	1.628	0.25
		(0.608–5.039)		(0.710–3.730)	
ASCT	Yes	0.409	0.04	0.405	0.01
		(0.175–0.954)		(0.201–0.814)	
*RAC1* mRNA	High	2.211	0.02	1.287	0.37
		(1.123–4.350)		(0.738–2.243)	

R-ISS, Revised-International Staging System; ASCT, autologous stem cell transplantation; OS, overall survival; PFS, progression-free survival; *RAC1*, Ras-related C3 botulinum toxin substrate 1; CI, confidence interval.

## Data Availability

All data that support the findings of this study are available from the corresponding authors upon reasonable request.

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
