# Peer review of "Role of Rac1 in p53-Related Proliferation and Drug Sensitivity in Multiple Myeloma"

_cancers, 2025, doi:10.3390/cancers17030461_

Round 1

Reviewer 1 Report (Previous Reviewer 2)

Comments and Suggestions for Authors

The updated manuscript titled " Role of Rac1 in p53-related proliferation and drug sensitivity in multiple myeloma " attempts at providing insights into Rac1 pathway and improving the prognosis of patients with multiple myeloma, including those with p53 deficiency and mutation.

There are following points which need careful attention:

1. In figure 6, please define “up” and “down”. Does author mean up-regulated /down-regulated genes? Currently it’s too generic for interpretation.

2. Please provide separate indent for test samples in figure 7 than embedding the labeling within the graph.

3.In figure 6, please improvise on annotating fig.A-D. Its very unclear.

4.Please check figure 4. Bars are cropped.

5. How do authors confirmed down regulation of RAC1 knockdown KMS11 and KMS26 cells using shRNA in Fig.4?

6. Bar graphs in Figure 3E is copped and what does author mean by “-“ and “+”? Also, please improvise Figure 3. Example Fig.3A-D,  It is currently mixed with survival and western images together. Consider further sub-sectioning it.

7. What does y axis in Fig. 3F represent?

8. Figure. 1 a says surviving cells but legend says proliferation rate. How is proliferation rate determined in this experiment?

Author Response

Reviewer 1

Thank you for reviewing our manuscript and we really appreciate your suggestion to improve our presentation.

  1. In figure 6, please define “up” and “down”. Does author mean up-regulated /down-regulated genes? Currently it’s too generic for interpretation.

Thank you for your comments. The “up” and “down” mean that the classification is based on significantly upregulated and downregulated genes by 1A-116 or ShRAC1, respectively. We changed the labels in Figure6.

  1. Please provide separate indent for test samples in figure 7 than embedding the labeling within the graph.

Thank you for your suggestions. We removed labels from the graphs and provided separate labels.

  1. In figure 6, please improvise on annotating fig.A-D. Its very unclear.

Thank you for your suggestions. We changed figure 6.

  1. Please check figure 4. Bars are cropped.

Thank you for your comments. We checked Figure 4, the CDKN1A mRNA expression levels were so low that the bar graph appears to be cropped but are not cropped.

  1. How do authors confirmed down regulation of RAC1 knockdown KMS11 and KMS26 cells using shRNA in Fig.4?

Thank you for your comments. We used reverse transcription quantitative PCR (RT-qPCR) to determine the decrease in RAC1 mRNA and confirmed the RAC1 knockdown.

Figure 4 has been modified for clarity the changes in RAC1 mRNA expression.

  1. Bar graphs in Figure 3E is copped and what does author mean by “-“ and “+”? Also, please improvise Figure 3. Example Fig.3A-D,  It is currently mixed with survival and western images together. Consider further sub-sectioning it.

Thank you for your suggestion. “-“ and “+” in Figure 3E mean HMCLs treated with 1A-116 0 µM and 50 µM, respectively. The label was changed to concentration.

The vertical labels in Figure 3E were changed probability to EdU positive cells.

In Figure 3, survival and western images were separated into sections.

  1. What does y axis in Fig. 3F represent?

Thank you for your comments. The Y axis in Figure 3F presents probability (%). We added labels to Figure.

  1. 1 a says surviving cells but legend says proliferation rate. How is proliferation rate determined in this experiment?

Thank you for your comments. Figure 1 shows surviving cells, so we modified the figure legend.

Reviewer 2 Report (New Reviewer)

Comments and Suggestions for Authors

This manuscript shows the role of Rac1 with p53 expression in multiple myeloma. Their results suggest that high RAC1 expression contributes to poor prognosis in multiple myeloma patients. The authors evaluated that in the experimental level and clinical level. Overall, the data are clearly presented. This reviewer has a few relatively minor comments, explained below.

Comments

1) The resolution of some graphs is low to understand the results in Fig 1A, Fig 2A-E, Fig 3A-F, Fig 4, Fig 5, and Fig 7. Mainly, the vertical labels are unclear. The authors need to refine the resolution, label description, and the size of character.

2) In Figure 5, it is difficult to understand what concentration conditions were taken and performed. It would be better to revise the visualization, e.g. using a log scale.

3) In Figure 6, the authors show the enrichment analyses of some cell lines. If the authors compare that, the multiple analysis mode of Metascape would be better to show the results.

4) Kaplan-Meier plots in Figure 7, the authors need to revise the lower panels showing the number at risk. It needs to be made clear what 0 and 1 represent.

5) in Line 205, there is a typo, RaC1. This reviewer recommends re-examining the text.

Author Response

Reviewer 2

Thank you for reviewing our manuscript and we really appreciate your suggestion to improve our presentation.

  1. The resolution of some graphs is low to understand the results in Fig 1A, Fig 2A-E, Fig 3A-F, Fig 4, Fig 5, and Fig 7. Mainly, the vertical labels are unclear. The authors need to refine the resolution, label description, and the size of character.

Thank you for your suggestion. We improved the resolution, label description, and the size of characters in Figures.

  1. In Figure 5, it is difficult to understand what concentration conditions were taken and performed. It would be better to revise the visualization, e.g. using a log scale.

Thank you for your suggestion. The X axis of the cereblon modulator concentration in Figure 5 was changed to a logarithmic scale.

  1. In Figure 6, the authors show the enrichment analyses of some cell lines. If the authors compare that, the multiple analysis mode of Metascape would be better to show the results.

Thank you for your suggestion. We used the multiple analysis mode of Metascape and changed Figure6.

  1. Kaplan-Meier plots in Figure 7, the authors need to revise the lower panels showing the number at risk. It needs to be made clear what 0 and 1 represent.

Thank you for your comments. 0 and 1 mean patients with low RAC1 mRNA expression and those with high RAC1 mRNA expression, respectively. We added labels to Figure 7.

  1. in Line 205, there is a typo, RaC1. This reviewer recommends re-examining the text.

Thank you for pointing out the mistake. We rechecked the text.

Additionally, gene names were unified in italics.

This manuscript is a resubmission of an earlier submission. The following is a list of the peer review reports and author responses from that submission.

Round 1

Reviewer 1 Report

Comments and Suggestions for Authors

Multiple RNA-seq analyses were performed, but meaningful validation should be done.

The mechanism of synergistic effects of the RAC1 inhibitor with CRBN modulators has not been fully analyzed.

Comments on the Quality of English Language

No comments in particular.

Reviewer 2 Report

Comments and Suggestions for Authors

The manuscript titled " Role of Rac1 in p53-related proliferation and drug sensitivity in multiple myeloma " provides insights into developing novel therapies targeting the Rac1 pathway and improving the prognosis of patients with multiple myeloma, including those with p53 deficiency and mutation.

Manuscript is interesting. However, several key revisions are warranted to enhance the clarity and impact of the manuscript:

1. Table 1 and 2 can either be part of paragraph in method and material section or as supplemental data. It is bit oddly placed as a part of main figure tables in current manuscript.

2. Please provide ethical consent statements/approvals for human samples.

3. Please provide details and statistical test utilized in GO analysis used in Fig.1C and Fig.6.

4. Statistical comparison among groups is missing in fig. 2e.

5. have authors analyzed IC50 value for RAC1 inhibitor 1A-116 in KMS11 and KMS26 cells? What is the criteria for 1A-116 -50 μM treatment to cells? Have authors examined dose dependent and time course dependent effect on apoptosis/p53 activation in these cells?

6. Please provide number of replicates for each immunoblot experiment.

7. Authors are encouraged to use “surviving cells” instead of “living cells” in graphs quantifying surviving cell population after inhibitor treatment.

8. Please provide number of replicates in fig.3e.

9. Please indicate the number of replicates in Figure 5 and specify the statistical test used. Are there any statistically significant differences among the multiple inhibitor-treated conditions? Additionally, the authors should improve the presentation of these graphs by clearly labeling the individual groups in the inset of each graph.